# Human Lanosterol 14-Alpha Demethylase (CYP51A1) Is a Putative Target for Natural Flavonoid Luteolin 7,3′-Disulfate

**DOI:** 10.3390/molecules26082237

**Published:** 2021-04-13

**Authors:** Leonid Kaluzhskiy, Pavel Ershov, Evgeniy Yablokov, Tatsiana Shkel, Irina Grabovec, Yuri Mezentsev, Oksana Gnedenko, Sergey Usanov, Polina Shabunya, Sviatlana Fatykhava, Alexander Popov, Aleksandr Artyukov, Olga Styshova, Andrei Gilep, Natallia Strushkevich, Alexis Ivanov

**Affiliations:** 1Institute of Biomedical Chemistry, 10 Building 8, Pogodinskaya Street, 119121 Moscow, Russia; pavel.ershov@ibmc.msk.ru (P.E.); evgeniy.yablokov@ibmc.msk.ru (E.Y.); yuri.mezentsev@ibmc.msk.ru (Y.M.); oksana@ibmh.msk.su (O.G.); alexei.ivanov@ibmc.msk.ru (A.I.); 2Institute of Bioorganic Chemistry NASB, 5 Building 2, V.F. Kuprevich Street, 220141 Minsk, Belarus; tvshkel@gmail.com (T.S.); grabovec-irina@mail.ru (I.G.); usanov@iboch.by (S.U.); iboh_lfhi@rambler.ru (P.S.); fsa1981@tut.by (S.F.); agilep@iboch.by (A.G.); 3G.B. Elyakov Pacific Institute of Bioorganic Chemistry, Far Eastern Branch of the Russian Academy of Science, 159 Prospect 100-letiya Vladivostoka, 690022 Vladivostok, Russia; popovam@piboc.dvo.ru (A.P.); artyukova@mail.ru (A.A.); krivoshapkoon@mail.ru (O.S.); 4Skolkovo Institute of Science and Technology, Bolshoy Boulevard 30, bld. 1, 121205 Moscow, Russia

**Keywords:** lanosterol 14-alpha demethylase, flavonoids, enzyme inhibition, surface plasmon resonance, spectral titration, molecular docking

## Abstract

Widespread pathologies such as atherosclerosis, metabolic syndrome and cancer are associated with dysregulation of sterol biosynthesis and metabolism. Cholesterol modulates the signaling pathways of neoplastic transformation and tumor progression. Lanosterol 14-alpha demethylase (cytochrome P450(51), CYP51A1) catalyzes one of the key steps in cholesterol biosynthesis. The fairly low somatic mutation frequency of CYP51A1, its druggability, as well as the possibility of interfering with cholesterol metabolism in cancer cells collectively suggest the clinical importance of CYP51A1. Here, we show that the natural flavonoid, luteolin 7,3′-disulfate, inhibits CYP51A1 activity. We also screened baicalein and luteolin, known to have antitumor activities and low toxicity, for their ability to interact with CYP51A1. The K_d_ values were estimated using both a surface plasmon resonance optical biosensor and spectral titration assays. Unexpectedly, in the enzymatic activity assays, only the water-soluble form of luteolin—luteolin 7,3′-disulfate—showed the ability to potently inhibit CYP51A1. Based on molecular docking, luteolin 7,3′-disulfate binding suggests blocking of the substrate access channel. However, an alternative site on the proximal surface where the redox partner binds cannot be excluded. Overall, flavonoids have the potential to inhibit the activity of human CYP51A1 and should be further explored for their cholesterol-lowering and anti-cancer activity.

## 1. Introduction

Cholesterol is a major source of bioactive sterols. Cholesterol modulates the signaling pathways of neoplastic transformation and tumor progression by covalently modifying hedgehog and smoothened family proteins [1,2] and it is also involved in atherosclerosis and metabolic syndrome progression [3]. Tumor growth is accompanied by a significant increase in the expression level of cholesterol biosynthetic enzymes, including lanosterol 14-alpha demethylase (cytochrome P450(51), CYP51A1) [4]. CYP51A1 belongs to the evolutionarily conserved family of cytochrome P450 and catalyzes the oxidative removal of the alpha-methyl group at the C14-position of the steroid substrate in three steps [5]. According to the COSMIC (http://cancer.sanger.ac.uk/cosmic accessed on 29 June 2020) resource, the CYP51A1 gene has a fairly low somatic mutation frequency (<0.1%) in various cancers. It has been shown that CYP51A1 gene expression correlates with the estrogen and progesterone receptor status of breast cancer [6] and could be one of the factors in assessing the survival rate of patients with gastric adenocarcinoma [7]. CYP51A1 catalyzes the production of 4,4-dimethyl-5α-cholesta-8,14,24-triene-3β-ol (follicular fluid meiosis-activating sterol, FF-MAS), one of the modulators of meiosis [8]. The ploidy disturbance characteristic for cancer cells is caused by processes similar to meiosis [9]. Thus, FF-MAS might be linked to the ploidy balance of tumor cells. CYP51A1 gene knockout blocked de novo cholesterol synthesis [10], while CYP51A1 inhibition led to the induction of apoptosis in cancer cells [11], indicating the clinical significance of this protein.

We analyzed the potential of natural flavonoids (Figure 1) as modulators of CYP51A1 function using purified human protein. Baicalein and luteolin are flavonoids that were originally isolated from plants of the *Scutellaria* and *Reseda* genus. The inhibitory activity of baicalein and luteolin was demonstrated for some cytochrome P450 isozymes [12,13,14]. Baicalein exhibited broad antifungal activity [15] and demonstrated strong synergy with fluconazole [16], a known inhibitor of fungal CYP51A1. Luteolin possesses an antibacterial effect in vivo, increasing membrane permeability [17], but does not directly perturb the model membranes in vitro [18,19]. Both baicalein and luteolin possess antitumor activity [20,21] and, as well as their derivatives, have been used in preclinical studies and in experimental oncology [22,23,24,25,26,27]. Epidemiological studies showed that foods rich in polyphenolic compounds (flavonoids, phenolic acids, lignans and stilbenes) included in the diet reduced the total risk of cancer by up to 50% [28]. Overall, 14 clinical trials were initiated to study baicalein and luteolin as dietary supplements (https://clinicaltrials.gov/, accessed on 1 March 2021). The G.B. Elyakov Pacific Institute of Bioorganic Chemistry has a broad collection of natural compounds from unique Far-Eastern plants and marine species with a wide range of biological activities. Luteolin 7,3′-disulfate, a water-soluble luteolin derivative originally obtained from the seagrass *Zostera marina* [29], also exhibits antitumor activity [30,31]. It was shown that sulfation at the 7-position of the luteolin molecule decreases cytotoxicity [32]. Moreover, the activity of luteolin 7,3′-disulfate in some cases is stronger than that of luteolin [30,33,34], possibly due to bypassing the stage of conjugation by intestinal and liver cells. Cholesterol is a crucial component of membranes, maintaining their permeability and fluidity. We hypothesized that flavonoids might target its synthesis via CYP51A1 and selected baicalein, luteolin and luteolin 7,3′-disulfate for testing. Using surface plasmon resonance (SPR), we showed that only luteolin 7,3′-disulfate interacted with CYP51A1 with high affinity. However, in the spectral binding experiments luteolin 7,3′-disulfate does not induce spectral changes. In contrast, baicalein and luteolin induce a reverse type I response in the difference absorption spectra of CYP51A1, indicating changes around heme iron. In the reconstituted enzymatic assay, among the three tested flavonoids, only luteolin 7,3′-disulfate inhibited the lanosterol 14α-demethylase activity of human CYP51A1 with significant potency. The binding mode distant from the heme was predicted for luteolin 7,3′-disulfate by the performed molecular docking, showing the binding not in the hydrophobic active site but rather in the access channel. The inhibitory effect of the most hydrophilic form of tested flavonoids—luteolin 7,3′-disulfate—is quite unusual. We suggest that, besides the predicted binding, luteolin 7,3′-disulfate could also bind to the proximal surface of CYP51A1, interfering with the interaction to the redox partner. The obtained data open up a new valuable source of flavonoid modulators of CYP51A1 activity as an alternative to the classic inhibition by azole compounds.

## 2. Results

### 2.1. Surface Plasmon Resonance

The CYP51A1 complex formation with flavonoids was detected using a SPR-biosensor. Lanosterol was used as a positive control to confirm the ability of immobilized CYP51A1 to bind ligands. With the CYP51A1 immobilized on the biosensor chip surface, we were able to detect the interaction with baicalein, luteolin and luteolin 7,3′-disulfate (Figure 2).

The equilibrium dissociation constant (K_d_) values of CYP51A1/flavonoid complexes were in the range of 2.9–20 μM, calculated association and dissociation rate constants are shown in Table 1. The obtained Kd value of the CYP51A1/lanosterol complex was 2.4 μM, which is comparable with the previously published data [35]. The association rate of the CYP51A1 complex with luteolin 7,3′-disulfate is seven times faster compared to the complex formation with lanosterol, while the dissociation rate is about eight times higher. The resulting K_d_ value for both complexes is similar. CYP51A1 complex formation with baicalein and luteolin is characterized by the increased association rate compared to lanosterol, but the main differences in the resulting K_d_ value are due to the great increase in dissociation rates of the complexes. Overall, the binding of flavonoids is faster compared to the natural substrate, but the dissociation of the complexes is faster as well. The highest affinity was detected for luteolin 7,3′-disulfate, which is more soluble.

### 2.2. Spectral Titration Analysis

The difference spectra of CYP51A1 were obtained by titration with baicalein, luteolin and luteolin 7,3′-disulfate in the presence of lanosterol. Baicalein and luteolin induced a reverse type I spectral response with absorbance minimum at 390 nm and maximum at 420 nm for luteolin and 436 nm for baicalein (Figure 3). These spectral changes are consistent with the previously detected interaction of cytochrome P450 1B1 with compounds of flavonoid class [36]. Titration with luteolin 7,3′-disulfate (up to 30 μM) does not cause changes in the difference spectrum of CYP51A1. The apparent dissociation constant (K_dapp_) values of the complexes of CYP51A1 with baicalein and luteolin were 8.2 ± 0.4 and 5.1 ± 0.5 μM, respectively. It should be noted that the K_d_ values from spectrophotometric titration experiments differ from those obtained using SPR. These differences can be attributed to the different affinities of the complexes in solution and immobilized on the surface of the optical chip. Interaction with the different sites of the enzyme cannot be excluded during SPR measurements and the measured Kd reflects all possible interactions between the ligand and enzyme, while spectral assays detect interactions of ligand only within close vicinity of the heme cofactor buried in the CYP active site.

### 2.3. Enzyme Activity Assay

Lanosterol 14α-demethylase activity of human CYP51A1 in the presence of flavonoids was determined in the reconstituted system. Only luteolin 7,3′-disulfate can inhibit the activity of the CYP51A1 (Table 2). Surprisingly, luteolin, being a more hydrophobic molecule compared to its sulfated derivative, does not have a similar effect. The apparent IC50 for luteolin 7,3′-disulfate is greater than 25 μM. At the same time, the level of inhibition by ketoconazole (94.6% at a concentration of compound of 5 μM) significantly exceeds the effect of luteolin 7,3′-disulfate (50.1% at a concentration of compound of 25 μM). Overall, the inhibition of CYP51A1 utilizing highly hydrophobic substrate by the water-soluble luteolin 7,3′-disulfate could not be predicted. This observation suggests a different mode of binding in the active site. To visualize the binding of luteolin and its disulfate in the active site we performed molecular docking.

### 2.4. Molecular Docking

We used a CYP51A1 crystal structure Protein Data Bank (PDB) ID: 3LD6 for molecular docking. The resulting models were selected based on the higher values of scoring function. The obtained docking poses are shown in Figure 4. Based on the docking results, luteolin binds very close to the heme coordinating iron (less than 3 Å) by the 3-OH-group of the phenyl ring. In contrast, luteolin 7,3′-disulfate binds at >8.5 Å from the heme. The docking results are consistent with the spectral titration data—luteolin induces reverse type I spectra, while luteolin 7,3′-disulfate does not change the spectral response.

Asp231 (C-terminal part of the F-helix) H-bonded to luteolin and is important for the enzymatic activity of CYP51 [37]. The negative charge in this position is highly conserved in Prokaryotes and Eukaryotes [38]. Residues Leu310 (part of I-helix), Met378 and Ile379 (both K-helix–β1-4 loop) are involved in the interaction with luteolin 7,3′-disulfate. Residues Leu310 and Met378 are conservative among *Chordata*, and Ile379 is conservative among primates [38]. Notably, these structural elements were shown to interact with the elongated azole inhibitors (PDB ID: 3LD6, 4UHI and 6Q2T), suggesting that several residues of the active site are utilized for the distant binding of luteolin 7,3′-disulfate.

The docking pose obtained for luteolin 7,3′-disulfate showed binding in the access channel (Figure 4). Thus, the inhibition effect could be the result of blocking of the substrate access channel. However, the inhibition of CYP51A1 by luteolin 7,3′-disulfate does not exclude the modulation of interaction with its redox partner. The proximal surface of CYP51A1—where the redox partner, cytochrome P450 reductase, is binding—contains positively charged amino acids which can interact with the negatively charged sulfate groups of luteolin 7,3′-disulfate.

## 3. Discussion

Plant flavonoids have a variety of biological activities in animals. However, despite numerous studies in this field, the mechanism/s of action of flavonoids remain poorly understood. Using animal models, it was shown that some flavonoids, luteolin in particular, may mitigate the toxicity of drugs [13,39]. However, the protective effect of flavonoids in humans has not been reliably ascertained [13]. There have been a number of studies reporting the effect of flavonoids, mostly on xenobiotic transformation by CYP enzymes and drug–drug interactions. In particular, baicalein showed an inhibition activity to CYP1A, CYP2B and CYP3A4, with IC50 values in the range of 0.5–36 μM [12,40]. Both baicalein and luteolin inhibit diclofenac 4′-hydroxylase activity in the CYP2C9 RECO system, with baicalein acting as a competitive inhibitor of CYP2C9 [41]. The most effective luteolin inhibition was shown for CYP2C8, while its close homologs, CYP2C9 and CYP2C19, were less effectively inhibited [13]. The activity of CYP1A2, CYP3A, CYP2B6, CYP2E1 and CYP2D6 was also inhibited by luteolin, with an IC50 in the range of 1.6–132.6 μM [13,42]. Notably, luteolin selectively inhibits CYP2D6-mediated metabolism with different substrates. For example, O-demethylation of 3-[2-(N,N-diethyl-N-methyl-ammonium)ethyl]-7-methoxy-4-methylcoumarin was inhibited to 40% by the administration of 20 μM luteolin, while the same concentration of luteolin showed less than a 5% inhibition in reaction with dextromethorphan [14]. Overall, the baicalein and luteolin inhibitory concentration on drug metabolizing CYPs is in the micromolar range.

The inhibition effect of flavonoids was also shown for CYPs involved in the biosynthesis of steroid hormones, neurosteroids, prostaglandins, as well as other regulatory metabolites. The effect of different flavonoids was evaluated on cortisol production in human adrenocortical H295R cells, and the competitive mechanism of inhibition was established for CYP21B1 [43]. The inhibition effect of luteolin was shown for human aromatase CYP19 [44]. A synthetic analogue of dihydrodaidzein, NV-52, inhibited the thromboxane A2 synthase CYP5A1 [45], while isoflavonoids inhibited the oxidation of vitamin D3 by CYP24A1 [46].

Sulfation, methylation and glucuronidation, occurring in the enterocytes and liver, are major factors affecting flavonoid bioavailability and are crucial for their transport via the blood [47]. Non-conjugated flavonoids are generally not present in plasma, however, there is an indication that a small amount of non-conjugated flavonoids can be transported through the blood system [48]. To the best of our knowledge, no studies have been conducted on the inhibition of CYP activity by sulfated forms of baicalein and luteolin. The inhibitory potential of other sulfated derivatives was probed with drug-metabolizing CYPs. It was shown that sulfated derivatives of quercetin and chrysin can inhibit several CYPs in vitro. In particular, quercetin 3′-sulfate has a selective concentration-dependent inhibition activity to CYP2C19 and CYP3A4 up to 30 μM, but the inhibition effect did not exceed 50% and overall was less than that of ticlopidine and ketoconazole (used as positive controls) [49]. Chrysin 7-sulfate has an IC50 value of 2.7 μM to CYP2C9, which is comparable to that of the positive control sulfaphenazole [50]. Additionally, chrysin 7-sulfate showed a slight inhibition effect on CYP2C19 and CYP3A4 [50].

CYP51A1 is considered as a potent target for cholesterol-lowering drugs [51]. There is an indication that the regulation of CYP51A1 function could be important to the treatment of oncological pathologies [11]. It was shown that anticancer drugs, abiraterone and galeterone, which are steroidal inhibitors of CYP17A1, can interact with human CYP51A1. However, their inhibition potential was not estimated. The Kdapp values determined for the abiraterone and galeterone, were 22 and 16 μM, respectively [52], and are significantly higher than those for baicalein and luteolin obtained in this work (8.2 and 5.1 μM, respectively). In contrast, non-steroidal pyridine derivative LK-935 [53] and azole inhibitors, ketoconazole and econazole [38], have an affinity in the submicromolar range due to direct coordination with heme iron. However, azole derivatives have a poor bioavailability and relatively low selectivity which might cause adverse reactions.

We showed the inhibition of CYP51A1 activity by the sulfated derivative of luteolin isolated from seagrass within the family *Zosteraceae* (*Zostera marina* and *Zostera asiatica)*. It was previously demonstrated that luteolin 7,3′-disulfate has a wide range of biological activities that might be linked to its higher bioavailability [33,54]. Considering that natural flavonoids and their biological activities are currently a subject of great interest and in light of our data, it is plausible to suggest that CYP51A1 activity could also be modulated by this group of compounds. Obtained data on inhibition by luteolin 7,3′-disulfate could be further explored for the development of a new class of CYP51A1 inhibitors.

## 4. Materials and Methods

### 4.1. Samples

Highly purified (>95% by SDS-PAGE) recombinant human CYP51A1 protein was expressed and purified as previously described [38]. Low molecular weight compounds: lanosterol (PubChem CID 246983, CAS Number 79-63-0), natural substrate of CYP51A1, and ketoconazole (PubChem CID 456201, CAS Number 65277-42-1), azole inhibitor of CYP51A1, were obtained from Cayman Chemicals (Ann Arbor, MI, USA), baicalein (PubChem CID 5281605, CAS Number 491-67-8) was obtained from Sigma Aldrich (St. Louis, MO, USA), luteolin (PubChem CID 5280445, CAS Number 491-70-3) and luteolin 7,3′-disulfate (PubChem CID 44258153) were purified in the G.B. Elyakov Pacific Institute of Bioorganic Chemistry (Vladivostok, Russia) by water–alcohol extraction, followed by chromatographic purification from the sea plants of *Zosteraceae* genus [55,56].

### 4.2. Surface Plasmon Resonance

SPR analyses were carried out at 25 °C using the optical biosensors Biacore T200 and Biacore 8K (GE Healthcare, Chicago, IL, USA) and sensor chips of CM5 series S type (Cytiva, Marlborough, MA, USA). HBS-N (10 mM HEPES, 150 mM NaCl, pH 7.4) (Cytiva) was used as a running buffer for CYP51A1 immobilization. Carboxyl groups of biosensor chip dextran were activated for 5 min by injection of the 1:1 mixture of 0.2 M 1-ethyl-3-(3-dimethylaminopropyl)carbodiimide hydrochloride (EDC) and 0.05 M N-hydroxysuccinimide (NHS) at a flow rate of 5 μL/min, followed by 1 min wash with HBS-N buffer at the same flow rate. Next, CYP51A1 (25 μg/mL) in 10 mM sodium acetate (pH 5.0) was injected into the working channel of the biosensor for 5 min at a flow rate of 5 μL/min. The final level of immobilization was 13,500 RU (13.5 ng of protein). Reference channel without immobilized CYP51A1 was used to correct the effects of the non-specific binding of analytes to the chip surface.

Baicalein and luteolin were prepared as 10 mM stock solutions in 100% dimethyl sulfoxide (DMSO). Experimental samples of baicalein and luteolin were prepared in an HBS-N buffer at the concentration range 10–100 μM and 1% DMSO. The same amount of solvent was added to the HBS-N running buffer to minimize bulk-effects introduced by the difference between the refractive indexes of the running buffer and the experimental samples. Refractive indexes of running buffer and experimental samples were matched with a precision refractometer RX-5000 (Atago, Saitama, Japan). If needed, the concentration of solvent in the running buffer was corrected according to the equation:$$C{\left(DMSO\right)}_{running\;buffer}=C{\left(DMSO\right)}_{sample}\times \frac{\eta_{1}-\eta_{2}}{\eta_{3}-\eta_{2}},$$
where *C(DMSO)_running buffer_*—DMSO final concentration in running buffer, *C(DMSO)_sample_* —DMSO concentration in experimental sample, *η*_1_—analyzed sample refractive index, *η*_2_—HBS-N buffer refractive index, *η*_3_—HBS-N buffer containing the DMSO of the same concentration as experimental sample refractive index.

Luteolin 7,3′-disulfate 10 mM stock solution and experimental samples at final concentrations of 10–100 μM were prepared with HBS-N buffer without organic solvent. The same buffer was used as a running buffer with luteolin 7,3′-disulfate to minimize the bulk-effects on the obtained experimental data. A total of 10 mM stock solution of lanosterol was prepared in ethanol. Lanosterol experimental samples at the final concentrations of 10–100 μM, as well as the running buffer, were prepared by the same protocol as for baicalein and luteolin but in ethanol instead of DMSO.

Low molecular weight compounds were injected through biosensor channels (working and reference) at a flow rate of 10 μL/min (luteolin 7,3′-disulfate) and 50 μL/min (baicalein and luteolin) for 6 min. Dissociation of the formed CYP51A1/compound complexes were registered at the same flow rate for no less than 6 min after the sample injection. After each biosensor cycle, a bound analyte was removed with two-times injection of regenerating solution (2 M NaCl, 1% CHAPS) at a flow rate of 30 μL/min for 30 s.

SPR sensorgrams were processed in Biacore T200 Evaluation Software v.1.0 (GE Healthcare) and BIAevaluation Software v 4.1.1 (GE Healthcare) using “1:1 (Langmuir) binding” and “Two-state (conformational change) binding” data processing models. The 1:1 (Langmuir) binding model is a model for the 1:1 interaction between compound (C) with immobilized protein (P), and is equivalent to the Langmuir isotherm for adsorption to a surface: C + P ↔ CP. Two-state (conformational change) binding model describes a 1:1 binding of compound (C) to immobilized protein (P) followed by a conformational change in the complex (CP ↔ CP*). It is assumed that the conformationally changed complex can dissociate only through the reverse of the conformational change: C + P ↔ CP ↔ CP*. The final kinetic parameters were obtained from the models with best fit of the experimental curves according to the minimum of the obtained chi^2^ value. The equations describing used models are as follows:

(1)1:1 (Langmuir) binding [57]: $K_{d}=\;\frac{k_{off}}{k_{on}}$, where *K_d_*—equilibrium dissociation constant, *k_off_*—dissociation rate constant, *k_on_*—association rate constant.(2)Two-state (conformational change) binding [58]: $K_{d}=\frac{k_{off1}}{k_{on1}}\times {\left(1+\frac{k_{on2}}{k_{off2}}\right)}^{-1}$, where *K_d_*—equilibrium dissociation constant, *k_off1_*—dissociation rate constant, *k_on1_*—association rate constant, *k_on2_*—forward rate constant for CP ↔ CP* transition, *k_off2_*—backward rate constant for CP ↔ CP* transition.

### 4.3. Spectral Titration Analysis

Spectrophotometric titration was used to determine the apparent dissociation constants (*K_dapp_*) for the enzyme–ligand complexes. The spectral measurements were performed on Cary Series UV-Vis-NIR (Agilent Technologies, Santa Clara, CA, USA) spectrophotometer using tandem quartz cuvettes (1 cm optical path) to exclude the absorption of ligands. Natural substrate, lanosterol, at final concentration 5 μM was added before the titration to the CYP51A1 (final concentration 4 μM) in 50 mM potassium phosphate buffer, pH 7.4. For titration, the ligand solution was added to the experimental cuvette (baicalein and luteolin 7,3′-disulfate were added up to a final concentration of 30 μM, luteolin was added up to a final concentration of 15 μM) and an equal volume of solvent was added to the control cuvette. The difference spectra were recorded after each addition of ligand at room temperature in the range of 350–500 nm. The apparent dissociation constants were determined by plotting the absorbance changes in the difference spectra versus the concentration of free ligand and evaluated by using the Hill equation (OriginPro 8.1 statistical data analysis package):$$A_{obs}=A_{max}\times \left(\frac{S\times n}{K_{dapp}\times n+S\times n}\right),$$
where *A_obs_*—the observed change in the absorption, *A_max_*—the absorbance change at ligand saturation, *K_dapp_*—the apparent dissociation constant for the ligand–enzyme complex, *S*—the ligand concentration, *n*—a Hill coefficient.

### 4.4. Enzyme Assay

Lanosterol 14-alpha-demethylase activity of human CYP51A1 was determined at 37 °C in 50 мM KPB, 4 mM MgCl_2_, 0.1 mM DTT in presence of lipids (0.15 мg/mL mixture 1:1:1 of L-α-dilauroyl-sn-glycero-3-phosphocholine, L-α-dioleoyl-sn-glycero-3-phosphocholine and L-α-phosphatidyl-L-serine). The final concentrations of CYP51A1 and CPR were 0.5 and 2.0 μM, respectively. Aliquots of concentrated recombinant proteins were mixed and preincubated for 5 min at room temperature. Lanosterol (10 mM stock solution in ethanol) was added to the reaction mixture at a final concentration of 50 µM. Tested compounds were added to the reaction mixture at a final concentration of 25 µM. To estimate the apparent IC50, the following concentrations of luteolin 7,3′-disulfate were used: 5, 10, 25, 50 and 100 μM. Ketoconazole at a concentration of 5 μM was used as a positive control. After 10 min of preincubation at 37 °C, the reaction was started by adding NADPH at a final concentration of 0.25 mM. Aliquots (0.5 mL) were taken from the incubation mixture at chosen time intervals. Steroids were extracted with 5 mL of ethyl acetate. The mixture was vigorously mixed, the water and organic phases were separated by centrifugation at 3000 rpm for 10 min. The organic layer was carefully removed and dried under argon flow. A total of 50 μL of methanol was added to the pellet and steroids were analyzed on a computerized Agilent 1200 series HPLC instrument (Agilent Technologies, USA) equipped with Agilent Triple Quad 6410 mass spectrometer (Agilent Technologies). Samples were analyzed by gradient elution on Zorbax Eclipse XDB C18 column (4.6 × 150 mm; 5 µm) (Agilent Technologies). A total of 0.1% (*v/v*) FA in water was used as the mobile phase A and 0.1% (*v/v*) FA in methanol:1-propanol mix (75:25, *v/v*) as mobile phase B. The gradient was 75–100% B in 0–5 min. The flow rate was 500 µL per min. The column temperature was maintained at 40 ± 1 °C. Mass spectrometry experiments were performed with atmospheric pressure chemical ionization source (APCI) at positive ion mode. The following APCI settings were used: gas temperature 200 °C, vaporizer 250 °C, gas flow 7 L/min, nebulizer pressure 40 psig, Vcap 4000 V, corona 4 µA, fragmentor 100 V. The data acquisition mode was MS2Scan from 200 to 550 Da.

### 4.5. Molecular Docking

Crystal structure of CYP51A1 PDB ID 3LD6 was used for docking. 3D structures of luteolin (CID 5280445) and luteolin 7,3′-disulfate (CID 44258153) were obtained from the PubChem database (https://pubchem.ncbi.nlm.nih.gov/, accessed on 7 November 2020). Removal of water and ligand molecules from the original protein PDB files and molecular docking over the entire surface of CYP51A1 were performed automatically in the Flare software package (Cresset, Litlington, UK) with default settings [59]. Docking hypotheses were arranged according to score functions values: Lead Finder (LF) Rank Score, LF dG, LF VSscore. The lower is the LF Rank Score, the higher is the likelihood that the docked pose reproduces the crystallographic pose. LF dG has been designed to perform accurate estimation of the free energy of protein–ligand binding for a given protein–ligand complex. LF VSscore has been designed to produce maximum efficiency in virtual screening experiments, i.e., to assign higher scores to active ligands (true binders) and lower scores to inactive ligands. Molecular graphics visualization tool Maestro version 12.5.139 (Schrödinger, New York, NY, USA) was used to analyze the selected docking hypotheses.

## 5. Conclusions

In this work, we identified a new ligand of human CYP51A1 among natural flavonoid—luteolin 7,3′-disulfate—that inhibits 14α-demethylase activity. Potential inhibitory mechanisms include blocking of either a substrate access channel or the interaction with a redox partner. Obtained results suggest further exploration of polyphenols for the cholesterol lowering ability and anti-cancer potential via CYP51A1.

## Figures and Tables

**Figure 1 molecules-26-02237-f001:**
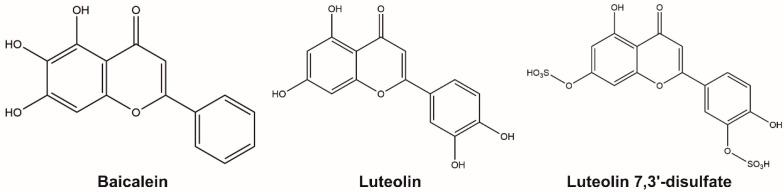
Flavonoids used in this work.

**Figure 2 molecules-26-02237-f002:**
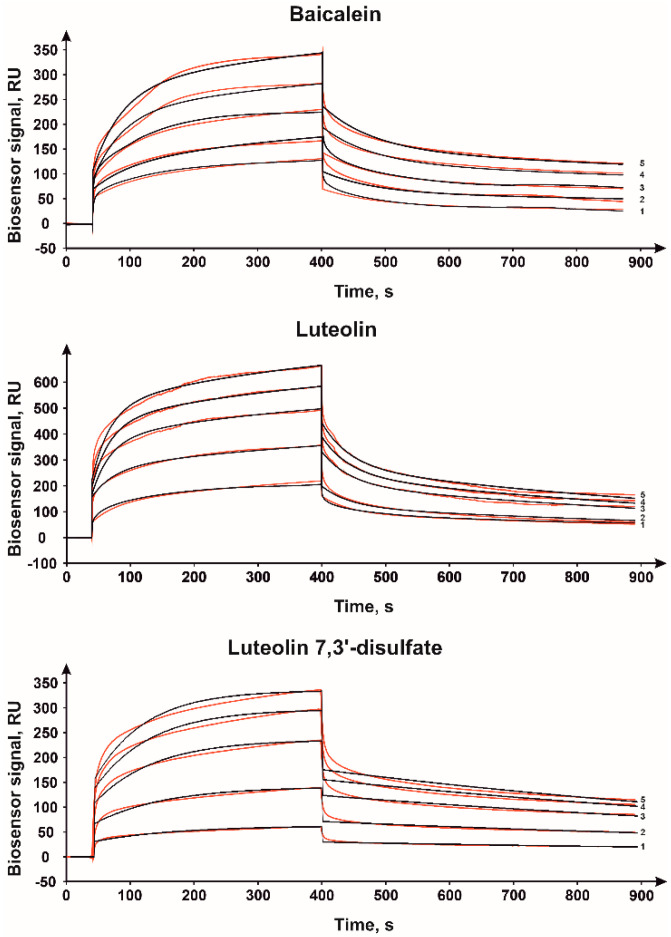
Typical surface plasmon resonance sensorgrams of binding between immobilized CYP51A1 on the optical chip and baicalein, luteolin and luteolin 7,3′-disulfate at different concentrations: 10 (1), 25 (2), 50 (3), 75 (4) and 100 μM (5). Fitting curves (theoretical models) are highlighted in black; Chi^2^ = 25.3 (baicalein), 68.2 (luteolin), 10.2 (luteolin 7,3′-disulfate).

**Figure 3 molecules-26-02237-f003:**
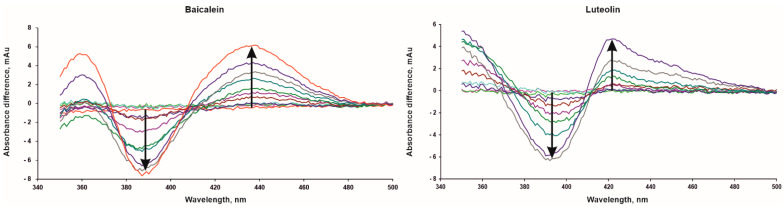
Difference spectra of CYP51A1 in the presence of lanosterol after addition of baicalein (up to 30 μM) and luteolin (up to 15 μM). The arrows indicate the direction of the spectral changes with increasing ligand concentration.

**Figure 4 molecules-26-02237-f004:**
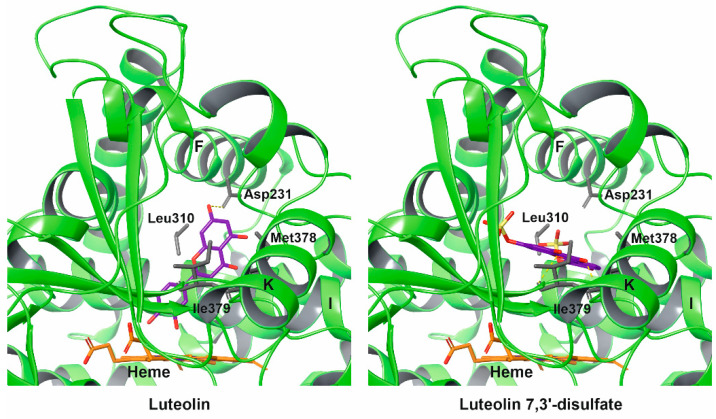
Luteolin and luteolin 7,3′-disulfate docked to the active site of human CYP51A1. The secondary structure of the protein is depicted as a ribbon and colored green. The amino acid side chains are shown as sticks and are colored in grey. The flavonoids and heme are shown as sticks and are colored in magenta and orange, respectively.

**Table 1 molecules-26-02237-t001:** Kinetic and equilibrium parameters of cytochrome P450(51) (CYP51A1) complex formation with lanosterol, baicalein, luteolin and luteolin 7,3′-disulfate.

Compound	k_on_	k_off_	K_d_, μM	Evaluation Model
lanosterol	k_on_ (1/Ms) = 41.4 ± 5.0	k_off_ (1/s × 10^−4^) = 1.0 ± 0.2	2.4	Langmuir 1:1
baicalein	k_on1_ (1/Ms) = 146 ± 20	k_off1_ (1/s × 10^−4^) = 100 ± 20	12.5	Two state reaction
	k_on2_ (1/s × 10^−4^) = 27 ± 3	k_off2_ (1/s × 10^−4^) = 6 ± 1		
luteolin	k_on1_ (1/Ms) = 282 ± 40	k_off1_ (1/s × 10^−4^) = 190 ± 30	20.0	Two state reaction
	k_on2_ (1/s × 10^−4^) = 33 ± 4	k_off2_ (1/s × 10^−4^) = 14 ± 2		
luteolin 7,3′-disulfate	k_on_ (1/Ms) = 294.0 ± 32.3	k_off_ (1/s × 10^−4^) = 8.4 ± 2.0	2.9	Langmuir 1:1

The table shows the average values of the parameters ± standard deviation, n = 3.

**Table 2 molecules-26-02237-t002:** Effect of compounds on catalytic activity of human CYP51A1 (lanosterol 14α-demethylase) in the reconstituted system in vitro.

Compound	Relative Activity, %
No compound	100.0
Baicalein (25 μM)	89.4
Luteolin (25 μM)	92.6
Luteolin 7,3′-disulfate (25 μM)	49.9
Ketoconazole (5 μM)	5.4

The final concentrations of CYP51A1 and cytochrome P450 reductase (CPR) were 0.5 and 2.0 μM, respectively. The final concentration of lanosterol was 50 μM.

## Data Availability

Datasets are available from the authors.

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
