# Peer review of "Human Lanosterol 14-Alpha Demethylase (CYP51A1) Is a Putative Target for Natural Flavonoid Luteolin 7,3′-Disulfate"

_molecules, 2021, doi:10.3390/molecules26082237_

Round 1

Reviewer 1 Report

Human 14-alpha demethylase (CYP51A1) is a putative target for anti-cancer flavonoids

Authors: Leonid Kaluzhskiy et.al

In this manuscript, the author Leonid Kaluzhskiy et.al., described Human sterol 14-alpha demethylase (CYP51A1) as a drug target for anticancer chemotherapy. In this study, the authors evaluated natural flavonoid, disulphate luteolin, inhibits CYP51A1 activity in the enzymatic activity assays.  

Overall, authors have well demonstrated disulphate luteolin as a CYP51A1 inhibitor using SPR optical biosensor and spectral titration assays. The experimental results and comparisons are clear and well demonstrated with SPR optical biosensor and spectral titration assays of disulphate luteolin. Therefore, the publication on Molecules

Author Response

The authors are grateful to you for your appreciation of their work.

Reviewer 2 Report

In this study, the interactions of luteolin-7,3’-disulfate, luteolin, and baicalein with CYP51A1 were examined. The study is interesting, however, some further experiments are required (mainly regarding the enzyme activity assay) to confirm the results. The manuscript has been written mostly in an appropriate scientific style; nevertheless, some improvements are strongly required. Therefore, I suggest to reconsider the manuscript only after a major revision. My critical comments are listed below.

Line numbering should be applied to make easier the evaluation of the manuscript.

The title is too general. Please, change it considering the experiments performed in the current study.

There are several typos and grammatical errors in the text. Please, check carefully and improve the manuscript.

Latin names (e.g. Scutellaria) should be written with italics.

"Cholesterol is one of the metabolic drivers of cancer…"

"Cholesterol is one of metabolic drivers of wide spread pathologies such as athero-sclerosis, metabolic syndrome and cancer."

As I know, the connection between cholesterol and cancer is a controversial issue (10.1158/0008-5472.CAN-15-2613). Therefore, a more careful wording in the corresponding parts is required. I know that the “anticancer” issue is very trending nowadays; however, there is no strong human evidence regarding the anticancer/antitumor effects of flavonoids. Because of the above-listed reasons, Authors should not emphasize so strongly this connection it in the whole manuscript.

“Human CYP51A1 is a recently emerging drug target for cholesterol lowering anti-cancer drugs.”

Could the Authors prove this statement with more, highly relevant references? If they do not, then clear this sentence.

“Moreover, the activity of luteolin 7,3'-disulphate in some cases is stronger than that of luteolin [29,32,33] and its bioavailability due to water-solubility is obviously more preferable.”

It is true that their poor aqueous solubility is one of the reasons of the low oral bioavailability regarding flavonoids. However, the hydrophilicity of sulfate conjugates can limit their transport by passive diffusion. Can the Authors support with relevant references their statement that luteolin-7,3'-disulphate has a better bioavailability than luteolin?

The caption of Table 2 does not include enough information about the experimental details.

I am not convinced based on the results described in the “Enzyme activity assay” section. Time- and concentration-dependence should be demonstrated. In addition, a widely accepted and known inhibitor of CYP51A1 should be also applied as a positive control. These details are indispensable to validate the method used.

Furthermore, IC50 or Ki values should be quantified.

An appropriate discussion of the results is almost completely missing.

Please, discuss your results considering the previously reported studies:

(1) comparison with other CYP51A1 inhibitors (e.g. 10.1124/dmd.111.040402; 10.1016/j.jinorgbio.2018.05.010)

(2) similar observations with other CYPs

(3) inhibition of CYPs by sulfate conjugates of flavonoids (chrysin, quercetin, etc.)

Author Response

  1. Reviewer 2 commentary:

Line numbering should be applied to make easier the evaluation of the manuscript.

Authors response:

In the manuscript submitted to the editorial office the line numbering was applied. Currently, the line numbering is removed but we will add it for convenience.

  1. Reviewer 2 commentary:

The title is too general. Please, change it considering the experiments performed in the current study.

Authors response:

The title “Human 14-alpha demethylase (CYP51A1) is a putative target for anti-cancer flavonoids” was changed to the:

“Human lanosterol 14-alpha demethylase (CYP51A1) is a putative target for natural flavonoid luteolin 7,3’-disulfate”

  1. Reviewer 2 commentary:

There are several typos and grammatical errors in the text. Please, check carefully and improve the manuscript.

Authors response:

Corrected accordingly. The term “disulphate” was changed to the “disulfate” due to IUPAC recommendations.

  1. Reviewer 2 commentary:

Latin names (e.g. Scutellaria) should be written with italics.

 Authors response:

Corrected accordingly.

  1. Reviewer 2 commentary:

"Cholesterol is one of the metabolic drivers of cancer…"

"Cholesterol is one of metabolic drivers of wide spread pathologies such as athero-sclerosis, metabolic syndrome and cancer."

As I know, the connection between cholesterol and cancer is a controversial issue (10.1158/0008-5472.CAN-15-2613). Therefore, a more careful wording in the corresponding parts is required. I know that the “anticancer” issue is very trending nowadays; however, there is no strong human evidence regarding the anticancer/antitumor effects of flavonoids. Because of the above-listed reasons, Authors should not emphasize so strongly this connection it in the whole manuscript.

Authors response:

We addressed this issue in both abstract (line № 18) and introduction (line № 39).

  1. Reviewer 2 commentary:

“Human CYP51A1 is a recently emerging drug target for cholesterol lowering anti-cancer drugs.”

Could the Authors prove this statement with more, highly relevant references? If they do not, then clear this sentence.

 Authors response:

We cleared this statement (line № 24).

  1. Reviewer 2 commentary:

“Moreover, the activity of luteolin 7,3'-disulphate in some cases is stronger than that of luteolin [29,32,33] and its bioavailability due to water-solubility is obviously more preferable.”

It is true that their poor aqueous solubility is one of the reasons of the low oral bioavailability regarding flavonoids. However, the hydrophilicity of sulfate conjugates can limit their transport by passive diffusion. Can the Authors support with relevant references their statement that luteolin-7,3'-disulphate has a better bioavailability than luteolin?

 Authors response:

We agree that passive transport of sulfated flavonoid derivatives is less effective than that of non-conjugated flavonoids [http://dx.doi.org/10.1016/j.bmc.2015.07.055]. However, there is an indication that sulfated flavonoids derivatives could be transported via organic anion transporters and organic anion transporting polypeptides [https://doi.org/10.1016/j.bcp.2012.05.011]. It is known that glucuronide or sulfate conjugates of luteolin can be found in plasma shortly after oral administration of the non-conjugated luteolin [https://doi.org/10.1021/jf062088r]. It is plausible to assume that luteolin 7,3’-disulfate could be transported through enterocytes directly to the plasma bypassing the stage of conjugation by intestinal and liver cells. It could also explain stronger effects of luteolin 7,3’-disulfate compared to the luteolin.

We modified the sentence to the following “Moreover, the activity of luteolin 7,3'-disulfate in some cases is stronger than that of luteolin [30,33,34], possibly due to bypassing the stage of conjugation by intestinal and liver cells.” (line № 78).

  1. Reviewer 2 commentary:

The caption of Table 2 does not include enough information about the experimental details.

  Authors response:

The Table 2 captionTable 2. Effect of compounds on catalytic activity of human CYP51A1 (lanosterol 14α-demethylase).” changed to the:

Table 2. Effect of compounds on catalytic activity of human CYP51A1 (lanosterol 14α-demethylase) in the reconstituted system in vitro.” (line № 162)

and

“The final concentrations of CYP51A1 and CPR were 0.5 and 2.0 μM, respectively. The final concentration of lanosterol was 50 μM.” (line № 163)

  1. Reviewer 2 commentary:

I am not convinced based on the results described in the “Enzyme activity assay” section. Time- and concentration-dependence should be demonstrated. In addition, a widely accepted and known inhibitor of CYP51A1 should be also applied as a positive control. These details are indispensable to validate the method used.

Furthermore, IC50 or Ki values should be quantified.

 Authors response:

We performed the additional experiments on concentration-dependent inhibition CYP51 by luteolin 7,3´-disulphate, following concentrations of luteolin 7,3'-disulfate were used - 5, 10, 25, 50 and 100 μM. However, due to the high variability of the data, we could not accurately determine the exact value of IC50 and estimated it as greater than 25 uM.

As a positive control we used ketoconazole. The information is included in the Results and Materials and Methods sections. (line № 154, 261, 355)

  1. Reviewer 2 commentary

An appropriate discussion of the results is almost completely missing.

Please, discuss your results considering the previously reported studies:

(1) comparison with other CYP51A1 inhibitors (e.g. 10.1124/dmd.111.040402; 10.1016/j.jinorgbio.2018.05.010)

(2) similar observations with other CYPs

(3) inhibition of CYPs by sulfate conjugates of flavonoids (chrysin, quercetin, etc.)

  Authors response:

The discussion is now included as a separate chapter and contains relevant information indicated by the reviewer. 

Reviewer 3 Report

The manuscript: "Human 14-alpha demethylase (CYP51A1) is a putative target for anti-cancer flavonoids" by Leonid Kaluzhskiy et al., contains valuable results that help the understanding of the potential value of flavonoids as anticancer agents. Nevertheless, it needs major revision before be accepted in Molecules Journal. 

Enzyme activity assay must incorporate more than one concentration of the flavonoids tested to have a reliable view of the inhibition activity reported. Furthermore, biochemical assays to determine the kind of inhibition must be presented to help to better understand the results presented in this manuscript.

The results and discussion section contains basically the results and just a poor discussion. This sections could be divided in order to offer to the readers a better understand and perspective.

The title could be more specific to the compounds tested and not as broad as "anticancer flavonoids". An alternative is to include in discussion section other flavonoids with the same structure features as those contained in the molecules tested.

Author Response

Reviewer 3

Comments and Suggestions for Authors

The manuscript: "Human 14-alpha demethylase (CYP51A1) is a putative target for anti-cancer flavonoids" by Leonid Kaluzhskiy et al., contains valuable results that help the understanding of the potential value of flavonoids as anticancer agents. Nevertheless, it needs major revision before be accepted in Molecules Journal.

  1. Reviewer 3 commentary:

Enzyme activity assay must incorporate more than one concentration of the flavonoids tested to have a reliable view of the inhibition activity reported.

 Authors response:

We performed the additional experiments on concentration-dependent inhibition CYP51 by luteolin 7,3´-disulfate and estimated IC50. As a positive control we used ketoconazole. The information is included in the Results and Materials and Methods sections. (line № 154, 261, 355)

  1. Reviewer 3 commentary:

Furthermore, biochemical assays to determine the kind of inhibition must be presented to help to better understand the results presented in this manuscript.

 Authors response:

To study the inhibition type of luteolin 7,3'-disulfate along with others flavonoid derivatives and their possible modulation of CYP51 interaction with redox partner is planned as separate follow up research.

  1. Reviewer 3 commentary:

The results and discussion section contains basically the results and just a poor discussion. This sections could be divided in order to offer to the readers a better understand and perspective.

 Authors response:

The result and discussion are now separated.

  1. Reviewer 3 commentary:

The title could be more specific to the compounds tested and not as broad as "anticancer flavonoids".

An alternative is to include in discussion section other flavonoids with the same structure features as those contained in the molecules tested.

 Authors response:

Corrected accordingly. The title “Human 14-alpha demethylase (CYP51A1) is a putative target for anti-cancer flavonoids” was changed to the “Human lanosterol 14-alpha demethylase (CYP51A1) is a putative target for natural flavonoid luteolin 7,3’-disulfate”.

Round 2

Reviewer 2 Report

Authors improved the manuscript. I accept their corrections and responses. I have no further critical comments.

Reviewer 3 Report

The revised manuscript has been greatly improved and now is acceptable for publication in Molecules Journal.